# The Efficacy of DW and T1-W MRI Combined with CT in the Preoperative Evaluation of Cholesteatoma

**DOI:** 10.3390/jpm12081349

**Published:** 2022-08-21

**Authors:** Wan-Hsuan Sun, Jiun-Kai Fan, Tzu-Chin Huang

**Affiliations:** 1Department of Otolaryngology, Tri-Service General Hospital, National Defense Medical Center, Taipei 114, Taiwan; 2Department of Radiology, Cathay General Hospital, Taipei 106, Taiwan; 3Department of Otolaryngology, Cathay General Hospital, Taipei 106, Taiwan

**Keywords:** combined DW-T1W-CT, cholesteatoma, TEES

## Abstract

Objective: This study aims to assess the efficacy of diffusion-weighted (DW) and T1-weighted (T1W) magnetic resonance imaging (MRI) combined with high-resolution computed tomography (HRCT) (together as DW-T1W-CT) in the preoperative evaluation of the presence and extent of cholesteatoma, which helps determine whether a patient is suitable for transcanal endoscopic ear surgery (TEES). Methods: This retrospective study included 35 patients (18 male and 17 female) aged from 2 to 81 years diagnosed with chronic otitis media with or without cholesteatoma, who had received surgical treatment and a preoperative MRI and HRCT during the period of December 2015 to December 2020 at Cathay General Hospital. We compared the preoperative DW-T1W-CT findings with the intraoperative findings and final pathologic diagnosis. The accurate predictive value was evaluated using the presence of cholesteatoma and its extent. Results: Regarding the efficacy of detecting cholesteatoma, we found a sensitivity of 92% (23/25 cases with cholesteatoma), a specificity of 90% (9/10 cases without cholesteatoma), and an overall accurate predictive value of 91.4% (32/35) by using combined DW-T1W-CT imaging. With regard to evaluating the extent of cholesteatoma, the combined DW-T1W-CT images obtained an accurate predictive value of 84% (21/25 cases of cholesteatoma). Conclusion: Combined DW-T1W-CT has been proven to be a reliable tool in detecting the presence of cholesteatoma. It is also useful in preoperatively depicting the extent of cholesteatoma, which is crucial for determining whether a patient is suitable for TEES, aiding in surgical planning and patient consultation.

## 1. Introduction

Mastoiditis with cholesteatoma is defined as a keratinized mass in the middle ear of a person with a history of head trauma, ear infection, or ear surgery. Cholesteatoma consists of keratinizing squamous epithelium. It can happen in the middle ear or mastoid cavity. Although cholesteatomas are not malignant tumors, they still can be invasive because of their erosive and expansible nature. Upper airway infection and otitis media could increase growth factors or matrix metalloproteinase, which make cholesteatoma more invasive. This can erode the surrounding bone in the middle ear and mastoid cavity, possibly furthering damage to the skull base or even to the brain [1,2]. Currently, microscopic ear surgery and endoscopic ear surgery are the main streams of the surgical treatment of cholesteatoma.

In the endoscopic ear surgery (EES) era, surgeons can rely much more on a precise assessment and depiction of ear disease preoperatively than on microscopic surgery, while such preoperative assessments also help them to determine the surgical technique and approach route to be employed. Transcanal endoscopic ear surgery (TEES) has been increasingly adopted to treat diseases of the middle ear, including cholesteatoma. TEES has a number of advantages, compared to microscopic ear surgery, in dealing with chronic otitis media (COM) with or without cholesteatoma [3].

In recent years, endoscopic ear surgery (EES) has been more broadly employed for managing COM with or without cholesteatoma. EES has many advantages, including a wider surgical view, more preservation of mastoid function, a smaller surgical wound, a shorter surgical time, and a shorter hospital stay compared to microscopic ear surgery [4]. However, this procedure still has some contraindications and limitations with regard to managing cholesteatoma, particularly concerning the anatomical structures, location, and extent involved [3,5,6]. In other words, understanding the characteristic and extent of disease prior to surgery is crucial for choosing the appropriate surgical technique and predicting surgical outcome. Some studies have suggested that cholesteatoma extending beyond the antrum and mastoid bone and pathologies involving the mastoid compartments (fistula of the labyrinth, fistula or dehiscence of the dura mater) are absolute contraindications to the exclusively transcanal endoscopic approach to cholesteatoma. Furthermore, related contraindications, such as stenosis, narrowing or exostosis of the external ear canal (EAC), and coagulopathies, have also been included [5]. Tarbachi et al. proposed that treating advanced cases involving the mastoid antrum could still be exteriorized through the “endoscopic open cavity management of cholesteatoma” [7]. As a result, estimating the existence and extent of cholesteatoma prior to surgery has become a crucial issue and is vital for determining the most appropriate surgical technique and plan.

Traditionally, high-resolution computed tomography (HRCT) of the temporal bone is the preferred preoperative imaging study, in general, and in the diagnosis of cholesteatoma, in particular.

HRCT can reveal the bony erosion caused by cholesteatoma with precise delineation of the soft tissue lesions in the temporal bone [1]. Nevertheless, HRCT cannot precisely differentiate cholesteatoma from granulation, fibrosis, and chronic inflammation opacity lesions of the tympanic cavity and mastoid [7,8]. 

For years, diffusion-weighted (DW) magnetic resonance imaging (MRI) has been used to evaluate middle ear cholesteatoma, primarily for detecting residual or recurrent cholesteatoma during postoperative follow-up [8,9,10,11,12]. Modern MRI techniques increasingly appear to be the imaging study of choice for the preoperative evaluation of cholesteatoma. By using diffusion-weighted magnetic resonance imaging (DW-MRI), cholesteatoma can be distinguished from surrounding inflammation tissue, granulations, and secretions, which can support the clinical diagnosis of cholesteatoma and help determine the disease extent more readily than CT scanning [7,9,10,11].

Our current protocol for preoperative evaluation of acquired and congenital cholesteatoma consists of a combination of HRCT and MRI, primarily based on non-EP DW and T1-weighted (T1W) images.

This retrospective study was conducted to evaluate the efficacy and accuracy of this combined DW-T1W-CT protocol in preoperatively assessing both the presence and extent of cholesteatoma.

## 2. Materials and Methods

### 2.1. Patients

This retrospective study included 35 patients (18 male and 17 female) aged from 2 to 81 years diagnosed with chronic otitis media with or without cholesteatoma, who had received surgical treatment and preoperative MRI and HRCT during the period of December 2015 to December 2020 at Cathay General Hospital. We excluded from this study those patients who underwent surgery but did not have complete preoperative MRI and HRCT imaging studies performed. This study was approved by the institutional review board (IRB number: CGH-P105012) and complied with the 1964 Helsinki Declaration and its later amendments or comparable ethical standards.

### 2.2. Image Study of the Temporal Bone

Preoperative evaluations included detailed history taking, a physical examination, and a hearing survey. The HRCT of the temporal bone is essential for preoperative evaluation of cholesteatoma and identifying the bony erosion of such middle ear structures as ossicles, fallopian canal, semicircular canals, and scutum. To obtain more precise and specific lesion detection, MRI was also taken concomitantly for all ear surgery candidates as part of the preoperative survey. MRI, which is mostly based on non-echo planar imaging (non-EPI) diffusion-weighted (DW) MRI and T1-weighted (T1W) images, can be used to distinguish cholesteatoma from other inflammation tissue, such as granulation, fibrosis, secretions, or cholesterol granuloma. Combining MRI images with the anatomical features in HRCT can further help depict the accurate range of cholesteatoma extension.

All the MRI and CT images were inspected and documented by experienced radiologists prior to surgery, who were blinded to the patients’ identities with regard to clinical and surgical findings.

For the convenience of documentation and analysis, the estimated extent of cholesteatoma based on combined DW-T1W-CT was classified into five categories: (0) no cholesteatoma; (1) involving the tympanic cavity proper; (2) confined in the attic; (3) extended to mastoid antrum; and (4) extended beyond the antrum (involving the peripheral mastoid air cell).

### 2.3. Surgical Techniques for Middle Ear Disease

In our department, the surgical approach for cholesteatoma primarily relies on the location and extent of the disease. For those cases w cholesteatoma that was confined within the tympanic cavity proper, attic, or mastoid antrum (categories 1 to 3), TEES is adopted, which includes transcanal, endoscopic tympanoplasty, atticotomy, or endoscopic open-cavity management of cholesteatoma, depending on the location and range of the cholesteatoma involved.

For more advanced cholesteatoma that extends beyond the mastoid antrum and involves the peripheral mastoid air cell, or for such complicated cases as those involving semicircular canal fistula, dehiscence of dura mater, and small external auditory canal, TEES is insufficient. Therefore, the microscopic trans-mastoid approach or endoscope assisted microscopic ear surgery are adopted for such extensive cases (category 4).

Intraoperative findings were recorded in detail, including the characteristic and involved region of the lesion, and the final diagnosis was confirmed by the pathological report.

### 2.4. Statistical Analysis

Assessing the presence of cholesteatoma

For all patients in the study group (35 cases), the preoperative DW-T1W-CT images were compared with intraoperative findings and pathologic diagnosis. We then assessed the sensitivity, specificity, and accurate predictive value with regard to predicting the presence of cholesteatoma. 

Assessing the extent of cholesteatoma

For those cases proven to be cholesteatoma after surgery (25 cholesteatoma cases), the preoperative DW-T1W-CT images and documents were reviewed and compared with intraoperative findings to evaluate the acute predictive value, with regard to determining the extent of cholesteatoma.

In this study, IMPAX Image Processing Software is applied to analyze CT and MRI image of patients with chronic otitis media. Validity statistical analysis is applied in analysis of image data.

## 3. Results

Of the 35 patients included in this study, 24 patients were diagnosed preoperatively as having otitis media with cholesteatoma, and 11 patients were diagnosed as having mastoiditis without cholesteatoma by using the combined DW-T1W-CT image study. Among those patients diagnosed as cholesteatoma, only one case was proven to be merely mastoiditis with granulation, but not cholesteatoma, after surgery.

Among those diagnosed with mastoiditis, two of them were actually found to be cholesteatoma cases after surgical and pathological inspection. One case was diagnosed as cholesterol granuloma, which is a special type of middle ear granulation tissue, as a presentation of mastoiditis (Table 1).

In predicting the presence of cholesteatoma, our study reveals a sensitivity of 92% (23/25 cases with cholesteatoma), specificity of 90% (9/10 cases without cholesteatoma), false positive of 4% (1/24 cases diagnosed with cholesteatoma but revealed negative with surgical finding), false negative of 18% (2/11 cases diagnosed without cholesteatoma but revealed positive with surgical finding) and an overall accurate predictive value of 91.4% (32/35 correct prediction), by way of preoperative combined DW-T1W-CT examination (Table 2). The false positive case (case 26) was originally diagnosed as otitis media with cholesteatoma, during the operation by the DW-T1W-CT images survey. However, the surgical finding revealed no cholesteatoma formation in the mastoid or middle ear cavity. (Figure 1) The false negative cases (case 24 and 25) were diagnosed as otitis media without cholesteatoma by the DW-T1W-CT images survey. Yet, during the operation, the surgical finding revealed cholesteatoma formation in the middle ear cavity or mastoid cavity (Figure 2 and Figure 3).

In this image series, CT showed mass lesion in middle ear, DWI MRI revealed hyper intensity signal and T1 MRI expressed mild peripheral ring enhancement around middle ear lesion. According to these features, otitis media with cholesteatoma is suspected in the pre-operative image evaluation. However, in the real situation, no cholesteatoma was found during the surgery or proved by the post-operative pathological diagnosis.

In this image series, CT showed mass lesion in middle ear cavity, intermediate-intensity DWI MRI signal and hyper-intensity T1 MRI signal were expressed in middle ear cavity. No definite evidence of cholesteatoma was noted in the pre-operative image evaluation. However, the cholesteatoma was revealed during the surgery as well as post-operative pathological examination.

In this image series, CT showed mass lesion in mastoid cavity, hypo-intensity DWI MRI signal and hyper-intensity T1 MRI signal were expressed in mastoid cavity lesion. No definite evidence of cholesteatoma was noted in the pre-operative image evaluation. However, the cholesteatoma was found during the surgery as well as post-operative pathological examination.

Furthermore, for the 25 cholesteatoma cases (including 3 congenital cholesteatoma and 2 recurrent cases), the surgical findings, including the location and invading range of cholesteatoma, were recorded and compared with the preoperative image studies. Our analysis reveals that the preoperative combined DW-T1W-CT image can have an accurate predictive value of 84% (21/25 correct prediction) regarding the extent of cholesteatoma (Table 3).

## 4. Discussion

In our study, the high validity of DW-T1W-CT in the preoperative evaluation of cholesteatoma is proven. Our study reveals a sensitivity of 92%, specificity of 90%, and an overall accurate predictive value of 91.4%, by way of preoperative combined DW-T1W-CT examination (Table 2).

High-resolution computed tomography (CT) continues to play a pivotal role in the preoperative evaluation of COM with or without cholesteatoma. CT scans can depict the anatomy of the middle ear and mastoid, as well as accurately predict the involvement of the sinus tympani and facial recess in the case of cholesteatoma [13]. Furthermore, CT scans have excellent spatial resolution, which allow for delineation of small soft tissue masses against bony structure, and can be a useful tool for diagnosing attic cholesteatoma [14]. This issue represents the main reason that diagnosing or excluding the presence of a cholesteatoma or predicting the disease extent based on CT findings alone is occasionally impossible. Aoki reported on the need in transmastoid antrotomy to achieve complete removal of the cholesteatoma in only 6 out of 24 patients, whose preoperative CT scans demonstrated shadowing of the epitympanum, antrum, and mastoid air cell [15]. Total resection of the cholesteatoma was possible by performing transcanal atticotomy combined with endoscope use in the other 18 patients. In these cases of partial or complete opacification of the middle ear and mastoid, when the cholesteatoma cannot be distinguished from the reactive changes but eradication by EES alone is possible, an MRI can provide essential information on the extension of the lesion in order to plan a minimally invasive transmeatal endoscopic approach [7].

Modern MRI techniques have increasingly emerged as the imaging study of choice in the preoperative evaluation of cholesteatoma and its postoperative follow-up. Using non-EPI DW-MRI, cholesteatoma can be distinguished from other tissue and mucosal retractions in the middle ear and mastoid [16]. The MRI characteristics of cholesteatoma are quite discriminative. On a standard T1-weighted MRI, both congenital and acquired cholesteatoma cases have a hypointense signal intensity, when compared with the brain’s gray matter. A cholesteatoma is a non-vascularized lesion and is, thus, not enhanced after intravenous gadolinium administration. In theory, after intravenous administration of gadolinium, the enhancement of the surrounding epithelial (matrix) and granular (perimatrix) layers can be seen as a thin, enhanced line (peripheral rim) on T1-weighted images. On non-EPI DW-MRI sequences, a cholesteatoma is characterized by a clear hyperintensity compared to the surrounding brain parenchyma (Figure 4, Table 4) [12].

With all these characteristics, an MRI has traditionally been considered the primary non-invasive method for excluding residual or recurrent cholesteatoma during postoperative follow-up, thus avoiding unnecessary second-stage surgery in most cases, especially in the pediatric population [9,12,17,18,19,20]. According to previous research, the sensitivity and specificity of DW-MRI detection of cholesteatomas have been reported as 91% to 94% and 92% to 96%, respectively [1,21,22,23]. Furthermore, performing T1-weighted magnetic resonance imaging (T1W-MRI) as an image exam to increase the specificity of diagnostic accuracy has been widely discussed [1,12]. False positive diagnosis of cholesteatoma from cholesterol granuloma, granulation tissue, abscess, and scar tissue was revealed in T1W-MRI images in several studies [1,24,25,26,27,28]. According to a study by Atsushi et al., compared to DW-MRI alone, combined T1W and DW-MRI may increase the overall specificity and diagnostic accuracy from 63.6% to 100% and from 87.7% to 91.2%, respectively [1].

A non-echo planar-based diffusion-weighted sequence (non-EP- DWI) has recently been described for evaluating middle ear cholesteatoma. This sequence has a higher resolution and thinner slice thickness (2 mm) and can show cholesteatoma as small as 2 mm, prior to primary and second-look surgery. A prospective evaluation of this non-EP-DWI sequence revealed a high sensitivity for detecting congenital or acquired middle ear cholesteatoma [10,12].

In the preoperative stage, the combined use of CT and MRI allows for unambiguous confirmation of the diagnosis, the ability to show the extent of the cholesteatoma, and evaluation of the risk for peri-operative sensory hearing loss related to labyrinthine fistulas, all of which can facilitate planning the surgical approach accordingly and better counseling the patient (Figure 1) [12].

Some limitations of DWI have been previously reported. Tiny congenital cholesteatoma, small retraction pockets, and evacuated cholesteatoma with an absence of keratin accumulation responsible for hyper intensity can be missed on an MRI [8,10,12]. Furthermore, reliable MRI interpretation requires a high level of training and experience for both the otologic surgeon and radiologist [12]. 

Our study reveals that the efficacy of detecting the presence of cholesteatoma using combined DW-T1W-CT image has a sensitivity of 92% (23/25 cases with cholesteatoma), a specificity of 90% (9/10 cases without cholesteatoma), and an overall accurate predictive value of 91.4% (32/35). Furthermore, comparing the preoperative assessments to the actual surgical findings demonstrated that the accurate predictive value of combined DW-T1W-CT images on the extent of cholesteatoma was 84% (21/25). Therefore, we believe that this tool is valuable for precisely predicting the presence and extent of cholesteatoma and can help determine the appropriate surgical procedure and plan for a patient’s best interests. Yet, our study also revealed a false positive rate of 4% and a false negative rate of 18% by using the combined DW-T1W-CT image method to detect cholesteatoma. Although this evaluating method is very accurate according to not only our study but also many other results from the studies mentioned above, preoperative discussion with patient including the thorough explanation of this minor uncertainty is crucial.

## 5. Conclusions

HRCT is currently the most widely used imaging tool for preoperatively diagnosing and characterizing cholesteatoma. However, the role of an MRI has become increasingly important. The more reliable information the surgeon can obtain prior to an operation, the better the surgical planning can be. The surgical outcome will also be enhanced when the otologist is flexible in choosing among the available surgical techniques and tailoring them to a patient’s clinical and imaging findings.

Combined DW-T1W-CT imaging has been proven to provide a reliable preoperative evaluation of diseases of the middle ear and can help precisely delineate the presence of cholesteatoma and its extension. It is a valuable tool for the otologic surgeon in determining the appropriate surgical technique and better planning, especially for candidates for transcanal endoscopic ear surgery.

## Figures and Tables

**Figure 1 jpm-12-01349-f001:**
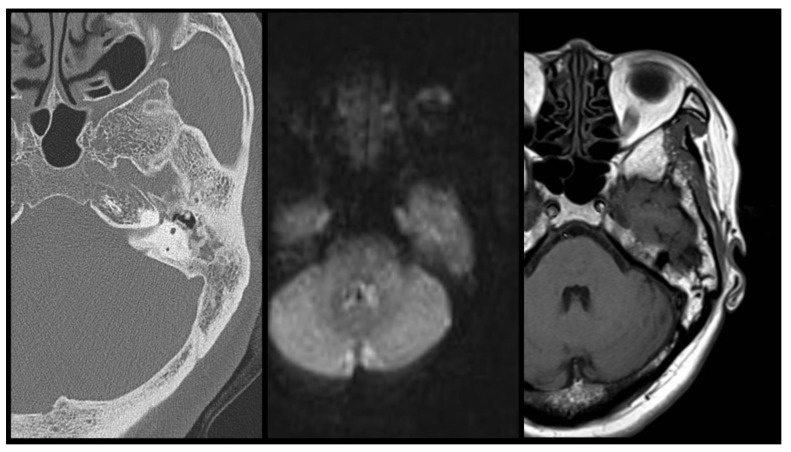
DW-T1W-CT images of false positive case (case 26). From left to right, CT-DWI-T1.

**Figure 2 jpm-12-01349-f002:**
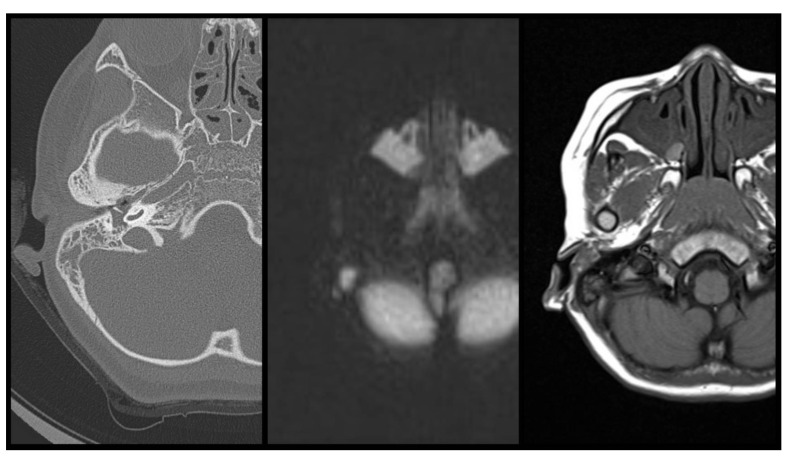
DW-T1W-CT images of false negative case (case 24). From left to right, CT-DWI-T1.

**Figure 3 jpm-12-01349-f003:**
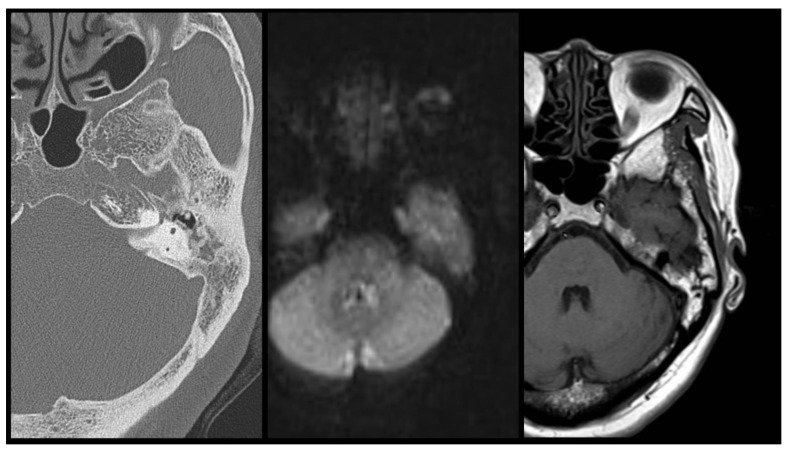
DW-T1W-CT images of false negative case (case 25). From left to right, CT-DWI-T1.

**Figure 4 jpm-12-01349-f004:**
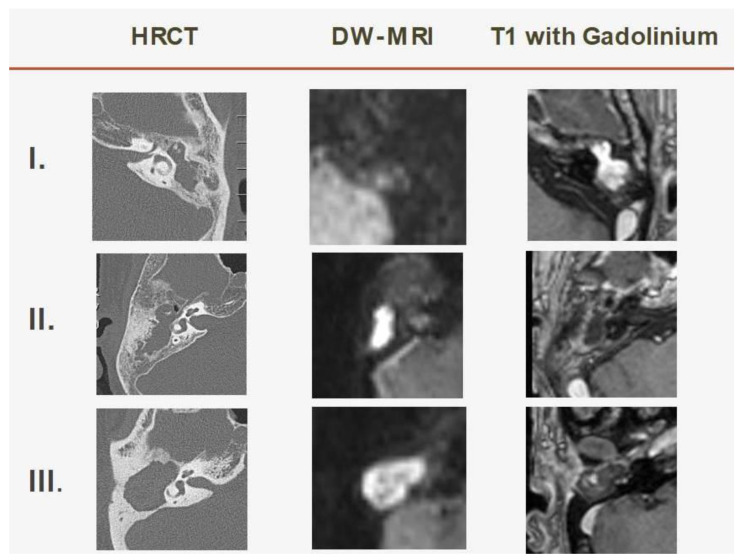
The assessment of cholesteatoma in mastoid cavity with combined DW-T1W-CT imaging. Case (**I**): Mastoiditis. The mastoid lesion has the characteristic of hypo-intensity on DW-MRI and hyper-intensity on T1W-MRI with gadolinium. Case (**II**): Cholesteatoma, confined in mastoid antrum. The mastoid lesion has the characteristic of hyper-intensity on DW-MRI and hypo-intensity and peripheral rim on T1W-MRI with gadolinium, with the extent limited within the mastoid antrum. Case (**III**): Cholesteatoma, extended beyond the mastoid antrum. The mastoid lesion has the characteristics of hyper-intensity on DW-MRI and hypo-intensity and peripheral rim on T1W-MRI with gadolinium, which extends beyond the mastoid antrum and involves the peripheral air cell, semicircular canal, and dura mater.

**Table 1 jpm-12-01349-t001:** Summary of cases regarding preoperative image assessment, diagnosis, and surgical findings.

Case	Age	Sex	Diagnosis of Disease		Extent of Cholesteatoma ****	
			Pre-OP Image	Post-OP Diagnosis	Pre-OP Image	Surgical Finding
1	64	F	Cholesteatoma	Cholesteatoma	2	4
2	49	M	Cholesteatoma	Cholesteatoma	4	4
3	53	M	Cholesteatoma	Cholesteatoma	1	1
4	35	M	Cholesteatoma	Cholesteatoma	3	3
5	41	M	Cholesteatoma	Cholesteatoma **	1	1
6	60	M	Cholesteatoma	Cholesteatoma	3	3
7	43	M	Cholesteatoma	Cholesteatoma	2	2
8	47	M	Cholesteatoma	Cholesteatoma	2	2
9	36	F	Cholesteatoma	Cholesteatoma	2	2
10	2	F	Cholesteatoma	Cholesteatoma *	1	1
11	61	F	Cholesteatoma	Cholesteatoma	3	3
12	38	F	Cholesteatoma	Cholesteatoma	2	2
14	8	M	Cholesteatoma	Cholesteatoma *	2	2
15	24	F	Cholesteatoma	Cholesteatoma	2	2
13	38	F	Cholesteatoma	Cholesteatoma	2	2
16	38	M	Cholesteatoma	Cholesteatoma **	2	2
17	27	F	Cholesteatoma	Cholesteatoma	2	2
18	81	F	Cholesteatoma	Cholesteatoma	4	4
19	63	M	Cholesteatoma	Cholesteatoma	3	3
20	47	F	Cholesteatoma	Cholesteatoma	1	2
21	19	M	Cholesteatoma	Cholesteatoma	3	3
22	61	M	Cholesteatoma	Cholesteatoma	4	4
23	41	M	Cholesteatoma	Cholesteatoma	3	3
24	8	F	Mastoiditis	Cholesteatoma *	0	1
25	52	F	Mastoiditis	Cholesteatoma	0	2
26	49	M	Cholesteatoma	Mastoiditis	1	0
27	52	F	Mastoiditis	Mastoiditis	0	0
28	55	F	Mastoiditis	Mastoiditis	0	0
29	59	F	Mastoiditis	Mastoiditis	0	0
30	48	F	Mastoiditis	Mastoiditis	0	0
31	31	M	Mastoiditis	Mastoiditis	0	0
32	57	F	Mastoiditis	Mastoiditis	0	0
33	44	M	Mastoiditis	Mastoiditis	0	0
34	49	M	Mastoiditis ***	Mastoiditis ***	0	0
35	21	M	Mastoiditis	Mastoiditis	0	0

(* congenital cholesteatoma; ** recurrent cholesteatoma; *** mastoiditis with cholesterol granuloma. **** Extent of cholesteatoma: 0: no cholesteatoma; 1: involving the tympanic cavity proper; 2: confined in attic; 3: extended to mastoid antrum; 4: extended beyond the antrum, involving peripheral mastoid air cells).

**Table 2 jpm-12-01349-t002:** The efficacy of preoperative DW-T1W-CT estimate on the presence of cholesteatoma.

Surgical Findings	Presence of Cholesteatoma	No Cholesteatoma
Pre-OP Image		
Presence of cholesteatoma	23	1
No cholesteatoma	2	9
Sensitivity	92% (23/25)	
Specificity		90% (9/10)
Accurate predictive value	91.4 (32/35)	

**Table 3 jpm-12-01349-t003:** The accuracy of preoperative DW-T1W-CT estimate on the extent of cholesteatoma.

Surgical Findings Pre-OP Image	Tympanic Cavity Proper	Attic	Antrum	Beyond Antrum
Not found	1	1		
Tympanic cavity proper	3	1		
Attic		9		1
Antrum			6	
Beyond antrum				3
Accurate predictive value	84% (21/25)			

**Table 4 jpm-12-01349-t004:** Differential diagnosis and imaging characteristics of cholesteatoma on MRI [7] .

	T1-Weighted MRI	T2-Weighted MRI	T1-Weighted MRI with Gadolinium	Diffusion-Weighted (DW) MRI
Cholesteatoma	Hypo-intensity	Hyper-intensity (intermediate to high)	Hypo-intensity (peripheral rim)	Hyper-intensity
Inflammation/ scar tissue	Hypo-intensity	Hyper-intensity	Hyper-intensity	No signal
Cholesterol granuloma	Hyper-intensity	Hyper-intensity	Hyper-intensity	Low-intermediate

## Data Availability

Research data supporting this publication are available by adequate request.

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
