# Peer review of "The Efficacy of DW and T1-W MRI Combined with CT in the Preoperative Evaluation of Cholesteatoma"

_jpm, 2022, doi:10.3390/jpm12081349_

Round 1
Reviewer 1 Report
The manuscript is well written, easy to understand and clearly explained. In my opinion, it can be accepted after minor revision addressing following comments:
- In the introduction more information about cholesteatoma and mastoiditis should be provided;
-The section "Result" should be "Results";
-In figure 1 a scheme showing the anatomical section represented in the panels should be inserted.
Author Response
Thank you for your valuable opinion.
-In the introduction more information about cholesteatoma and mastoiditis should be provided;
We have added more information about cholesteatoma and mastoiditis in the first paragraphof "Introduction" :
Mastoiditis with cholesteatoma is defined as keratinized mass in the middle ear with history of head trauma, ear infection or ear surgery. Cholesteatoma consist of keratinizing squamous epithelium. It can happen in middle ear or mastoid cavity. Although cholesteatomas are not malignant tumors, they still can be invasive because of their erosive and expansible nature. Upper airway infection and otitis media could increase growth factors or matrix metalloproteinase which make cholesteatoma more invasive. This can erode surrounding bone in the middle ear and mastoid cavity, further possibly damage to the skull base or even to the brain. Currently, microscopic ear surgery and endoscopic ear surgery are main streams of surgical treatment of Cholesteatoma.
-The section "Result" should be "Results";
We have corrected "Result" to "Results".
-In figure 1 a scheme showing the anatomical section represented in the panels should be inserted.
In figure 1, all the panels describe the disease in mastoid cavity. Therefore, we added the anatomic description of thise region in the caption of figure 1 as following:
The assessment of cholesteatoma in mastoid cavity with combined DW-T1W-CT imaging.
Reviewer 2 Report
This study analyzed the imaging assessment of DWI in MRI and CT in the cholesteatoma, and showed the efficacy of DWI. Although this theme is not novel, the data may be meaningful to clinicians specialized in otological surgery.
> Does the introduction provide sufficient background and include all relevant references?
The first and the part of second paragraph in the discussion section should be in the introduction section. Some other parts in the discussion section will be in the introduction section.
The first paragraph of the discussion section should briefly summarize the results of the study.
> Are the methods adequately described?
The name of the software and the type of the statistical analysis should be stated.
> Are the results clearly presented?
Because the false positive of the DWI includes Cholesterin Granuloma, Granuloma should be also stated clearly rather than in the legend section.
Because the theme is not novel, the authors should present false-positive or false-negative cases (of all cases if possible) in the article to make the article more valuable.
In addition, the discussion on the false-positive or false-negative cases in this study should be added rather than just presenting the specificity or sensitivity of previous reports.
Author Response
Thank you very much for these important suggestion. Our answers is as following:
> Does the introduction provide sufficient background and include all relevant references?
The first and the part of second paragraph in the discussion section should be in the introduction section. Some other parts in the discussion section will be in the introduction section.
The first paragraph of the discussion section should briefly summarize the results of the study.
Answer:
We have changed first parahraph of "Discussion" section to third paragraph of "Introduction" section.
We also added brief summary of study result in the first paragraph of "Discussion" section as following:
In our study, high validity of DW-T1W-CT in preoperative evaluation of cholesteatoma is proved. Our study reveals a sensitivity of 92%, specificity of 90%, and an overall accurate predictive value of 91.4% by way of preoperative combined DW-T1W-CT examination (Table 2).
> Are the methods adequately described?
The name of the software and the type of the statistical analysis should be stated.
Answer:
We have added the detail of analysis information in the last paragraph of "Materials and Methods" section as following:
In this study, IMPAX Image Processing Software is applied to analyse CT and MRI image of patients with chronic otitis media. Validity statistical analysis is applied in analysis of image data.
> Are the results clearly presented?
Because the false positive of the DWI includes Cholesterin Granuloma, Granuloma should be also stated clearly rather than in the legend section.
Answer:
We added more information about cholesterol granuloma, to make readers understand clearly about this disease which is catagorized as mastoiditis but not cholesteatoma. Description as following:
Among those diagnosed with mastoiditis, two of them were actually found to be cholesteatoma cases after surgical and pathological inspection. One case was diagnosed as cholesterol granuloma which is special type of middle ear granulation tissue as a presentation of mastoiditis. (Table 1)
Because the theme is not novel, the authors should present false-positive or false-negative cases (of all cases if possible) in the article to make the article more valuable.
In addition, the discussion on the false-positive or false-negative cases in this study should be added rather than just presenting the specificity or sensitivity of previous reports.
Answer:
We added the information of false positive and false negative in the result paragraph as following:
In predicting the presence of cholesteatoma, our study reveals a sensitivity of 92% (23/25 cases with cholesteatoma), specificity of 90% (9/10 cases without cholesteatoma), false positive of 4% (1/24 cases diagnosed with cholesteatoma but revealed negative with surgical finding) , false negative of 18% (2/11 cases diagnosed without cholesteatoma but revealed positive with surgical finding) and an overall accurate predictive value of 91.4% (32/35 correct prediction) by way of preoperative combined DW-T1W-CT examination (Table 2).
Also, we added more novel opinion in the last paragraph of discussion to emphasize the meaning of out false positive and false negative results as follwing:
Yet, our study also revealed false positive rate of 4% and false negative of 18% by using combined DW-T1W-CT image method to detect cholesteatoma. Although this evaluating method is very accurate according to not only our study but also many other results from studies mention above, preoperative discussion with patient including thorough explanation of this minor uncertainty is crucial.
Reviewer 3 Report
The authors analyzed pre-op evaluation of chronic middle ear disease in 25 patients with cholesteatoma and 10 pts with other disease. The results show 23/25 correctly diagnosed cholestoma and 9/10 non-cholesteatomas correctly identified. 21/25 cholesteatomas were correctly described with the two radiology methods combined (MR diffusion weighting and hi-res CT scan) in order to pre-op be able to decide on the surgical approach (classic matoidectomy or endoscopy via the ear canal). This is clinically relevant and the authors' methods and analysis is well done in my opinion. The radiology images are also good.
Author Response
Thank you for your valuable opinion.
Round 2
Reviewer 2 Report
Most concerns were fixed in the revised version.
Is it possible to show the specific CT/MRI images of the false-negative and the false-positive cases?
Author Response
-Is it possible to show the specific CT/MRI images of the false-negative and the false-positive cases?
Thank you very much for the valuable suggestion. We have added three figures of one false positive case and two false negative cases and added the content in "Results" paragraph as following:
False positive case (case 26) original diagnosed as otitis media with cholesteatoma, during operation by DW-T1W-CT images survey. But surgical finding revealed no cholesteatoma formation in mastoid or middle ear cavity. (Fig 1) False negative cases (case 24 and 25) diagnosed as otitis media without cholesteatoma by DW-T1W-CT images survey. Yet during operation, surgical finding revealed cholesteatoma formation in middle ear cavity ormastoid cavity. (Fig 2, 3)
Figure 1 caption:
Figure 1. DW-T1W-CT images of false positive case (case 26). (From left to right, CT-DWI-T1) .
In this image series, CT showed mass lesion in middle ear, DWI MRI revealed hyper intensity signal and T1 MRI expressed mild peripheral ring enhancement around middle ear lesion. Original diagnosed as otitis media with cholesteatoma by image study. But during operation, surgical finding revealed no cholesteatoma formation in mastoid or middle ear cavity.
Figure 2 caption:
Figure 2. DW-T1W-CT images of false negative case (case 24). (From left to right, CT-DWI-T1)
In this image series, CT showed mass lesion in middle ear cavity, iso-intensity DWI MRI signal and hyper-intensity T1 MRI signal were expressed in middle ear cavity. Original diagnosed as otitis media with no cholesteatoma by image study. Yet during operation, surgical finding revealed cholesteatoma formation in middle ear cavity.
Figure 3 caption:
Figure 3. DW-T1W-CT images of false negative case (case 25). (From left to right, CT-DWI-T1)
In this image series, CT showed mass lesion in mastoid cavity, hypo-intensity DWI MRI signal and hyper-intensity T1 MRI signal were expressed in mastoid cavity lesion. Original diagnosed as otitis media with no cholesteatoma by image study. Yet during operation, surgical finding revealed cholesteatoma formation in attic cavity.